# Potential Role of Low-Molecular-Weight Dioxolanes as Adjuvants for Glyphosate-Based Herbicides Using Photosystem II as an Early Post-Treatment Determinant

**DOI:** 10.3390/cells12050777

**Published:** 2023-02-28

**Authors:** Ewa Szwajczak, Edyta Sierka, Michał Ludynia

**Affiliations:** Institute of Biology, Biotechnology and Environmental Protection, University of Silesia in Katowice, 40-032 Katowice, Poland

**Keywords:** herbicide, chlorophyll fluorescence, adjuvants, glyphosate, weed control

## Abstract

Pesticide use cannot be completely abandoned in modern agriculture. Among agrochemicals, glyphosate is one of the most popular and, at the same time, most divisive herbicide. Since the chemicalization of agriculture is detrimental, various attempts are being made to reduce it. Adjuvants—substances that increase the efficiency of foliar application—can be used to reduce the amount of herbicides used. We propose low-molecular-weight dioxolanes as adjuvants for herbicides. These compounds quickly convert to carbon dioxide and water and do not harm plants. The aim of this study was to evaluate the efficacy of RoundUp^®^ 360 Plus supported by three potential adjuvants: 2,2-dimethyl-1,3-dioxolane (DMD), 2,2,4-trimethyl-1,3-dioxolane (TMD), and (2,2-dimethyl-1,3-dioxan-4-yl)methanol (DDM), on a common weed species *Chenopodium album* L., under greenhouse conditions. Chlorophyll *a* fluorescence parameters and analysis of the polyphasic fluorescence (OJIP) curve, which examines changes in the photochemical efficiency of photosystem II, were used to measure plant sensitivity to glyphosate stress and verified the efficacy achieved by tested formulations. The effective dose (ED) values obtained showed that the weed tested was sensitive to reduced doses of glyphosate, with 720 mg/L needed to achieve 100% effectiveness. Compared to the glyphosate assisted with DMD, TMD, and DDM, ED was reduced by 40%, 50%, and 40%, respectively. The application of all dioxolanes at a concentration equal to 1 vol.% significantly enhanced the herbicide’s effect. Our study showed that for *C. album* there was a correlation between the change in OJIP curve kinetics and the applied dose of glyphosate. By analyzing the discrepancies in the curves, it is possible to show the effect of different herbicide formulations with or without dioxolanes at an early stage of its action, thus minimizing the time for testing new substances as adjuvants.

## 1. Introduction

Land cultivation has always played an essential role in the history of mankind. It is a significant source of food, fodder and raw materials for industry. The increase in production in the last century would not have been possible without improvements in the efficiency and productivity of cultivation. Efforts to increase yields have yielded results mainly through introducing high-yielding varieties and using chemicals such as fertilizers and pesticides [1,2,3]. In addition to fertilizers, pesticides seem to be the most effective solution for farmers. To maximize yields, most crops are grown in monocultures, ecosystems in which no plants other than crops are allowed to grow, so every effort is made to maintain this condition [4]. To this end, herbicides have been developed to protect crops and increase yield and quality, and remain the most effective, efficient, and environmentally friendly method of weed control [5,6].

In the past century, weed control methods have changed from mechanical and mechanochemical to almost exclusively chemical [6,7]. For the environment, the active ingredients in herbicide formulations are xenobiotic pollutants that can negatively or directly affect various organisms and disrupt their physiological processes, reproduction, or behavior. Herbicides’ most apparent side effects are minor or significant disturbances in plant growth and metabolism [4]. In crops, they have been observed to disrupt growth, reduce biomass, and impair germination and flower and seed production. Such effects were observed in the crop where the herbicide was applied and subsequent sowings [8,9,10]. The risks posed by herbicides arise from their persistence in the environment and the possibility of spreading beyond the application area [9,11]. Due to the growing awareness of the negative impact of pesticides on the environment, intensified agriculture has begun to make efforts to reduce both the dosage and the number of treatments with pesticides, especially herbicides, to control weeds without reducing the quantity and quality of the crop [12,13,14].

However, many farmers still believe that eliminating agrochemicals and completely replacing the use of synthetic pesticides with alternative farming methods could lead to yield losses [15]. One way to improve the efficacy of herbicides while reducing the dose is to increase the contact of the product with the plant surface and improve the absorption capacity of these agents into the cells. Absorption into plant tissue is a basic requirement for the efficacy of foliar-applied herbicides. However, since the aerial parts of most plants are not adapted to absorb water and compounds dissolved in water, this process is complex and much more challenging to manage than in the roots [16,17,18,19].

Factors primarily related to plant morphological and anatomical characteristics influence the uptake of foliar-applied herbicides. For example, the epidermis forms the first barrier to the penetration of many xenobiotic substances into the plant. The outer periclinal secondary cell wall of the epidermis is thicker and covered with a multilayered cuticle. This structure plays a crucial, if not the most critical, role in the penetration of herbicides, as it can limit their retention on the leaf surface and efficient diffusion into the cells [17,20,21].

Since the development of land cultivation, attempts have been made to enhance the effects of agrochemicals by using various additives. For most foliar-applied herbicides, it is recommended to use them with an appropriate adjuvant [17,18,19,20,21,22]. Adjuvants are defined as ingredients that enhance or modify the action of the active ingredient or improve the physicochemical properties of the formulation, allow stability and shelf life of the formulation during storage, minimize problems in spray preparation, or even prevent drift [17,23,24]. Adjuvants have been developed primarily for foliar herbicides to promote surface contact of the active ingredient, increase coverage, retain droplets, and delay evaporation of the liquid in which they are dissolved to then remain on the plant surface and penetrate plant tissue [25,26,27]. The efficacy of formulations containing significantly lower levels of active ingredients and adjuvants has been demonstrated under field conditions for several adjuvants. For example, the use of oil-based adjuvants and nonionic surfactants reduced the herbicide dose by 25–50% [13,28,29,30].

Faced with dwindling fossil fuel resources and the need to reduce greenhouse gas emissions, the agrochemical industry is increasingly opting to use so-called “bio” solutions that simultaneously meet sustainable development commitments [6]. Cyclic acetals, as well as other glycerol-derived chemicals [31,32], are used for their properties as surfactants, solvents in paints and coatings, bioadditives for fuels, to reduce emissions of carbon monoxide, hydrocarbons, and aldehydes, or in controlled release systems for drugs or pesticides [32,33,34]. The structure consists of monomers of five or more units, at least one of which contains two oxygen atoms flanking an unsubstituted or substituted methylene group. Among these, 1,3-dioxolanes have been widely recognized in the industry but have only recently been described as non-toxic bio-derived chemicals. Three of them, 2,2-dimethyl-1,3-dioxolane (DMD), 2,2,4-trimethyl-1,3-dioxolane (TMD), and (2,2-dimethyl-1,3-dioxan-4-yl)methanol (DDM) (Figure 1), have recently been analyzed for agricultural applications [35].

These 1,3-dioxolanes have only two or three simple substituents attached to the core of the compound. Fleute-Schlachter et al. [36] also described compounds based on the structure of 1,3-dioxolane in their patent. They suggested a wide agricultural application of alkoxylated glycerol acetals with different hydrocarbon residues with 6 to 22 carbon atoms, e.g., together with herbicides, insecticides, fungicides, or plant growth promoters because they improve the ability of the active ingredients to remain on the surface and penetrate deeply into the target organism. The same use is suggested by Kapkowski et al. [36] for DMD, TMD, and DDM as additives to herbicides or foliar fertilizers to improve their uptake. Moreover, Kapkowski et al. [37] described in their patent the application as non-toxic and easily degradable by plants. The authors also reported the positive uptake of indolyl-3-acetic acid (IAA), and naproxen into the plant via foliar uptake induced by dioxolanes and their use to transport simple compounds into the plant. Based on these properties, they can be classified as adjuvants.

DDM, also known as solketal, has a high affinity for both polar and nonpolar substances due to the presence of two chemical groups of different polarities (a cyclic ether group and a hydroxyl group) [31]. Presumably, the degree of absorption of the substances applied to the leaves depends on the surface tension and the wetting angle of the solutions in which they were dissolved. DDM, DMD, and TMD have lower values for surface tension and wetting angle compared to the properties of water. Therefore, the addition of 1,3-dioxolanes to aqueous solutions results in better droplet retention and more effective wettability of leaf surfaces [34]. The second parameter, logP (describing lipophilicity), is also related to the penetration of xenobiotics through biological barriers. Due to the lipophilic nature of the cuticle, the ability of compounds to penetrate leaves rises with increasing lipophilicity [16,34]. In the case of 1,3-dioxolanes, logP > 0 values are favorable and, together with lower surface tension and wetting angle, may increase the absorption of compounds by the cuticle.

This study aims to evaluate the efficacy of 1,3-dioxolanes as adjuvants for herbicides by assessing the efficiency of the photosynthetic apparatus in the early phase of action. This complex system of biochemical reactions that converts light energy into chemical energy can be used to evaluate the efficacy of herbicides, even though they do not actively affect photosynthesis [38,39]. In this evaluation, the question is whether such a physiological indicator changes under the stress induced by Roundup^®^ 360 Plus herbicide alone and in combination with 1,3-dioxolane. The experiment’s main objective is to determine whether adding a potential adjuvant is more effective against *Chenopodium album* by analyzing chlorophyll *a* fluorescence (ChlF) a long time before visible symptoms. This method is beneficial for testing new adjuvants that can be used to improve herbicide efficacy.

## 2. Materials and Methods

### 2.1. Plant Material

*Chenopodium album* L., also known as lamb’s quarters or goosefoot, belongs to the *Amaranthaceae* family. It is a common dicotyledonous weed that grows in fields and ruderal habitats. This annual spring plant is one of the five most widespread weeds in the world, especially in Europe. This weed competes with crops for nutrients, light, and water, resulting in lower yields of sugar beet, barley, mustard, corn, foxtail millet, and many other crops [40,41,42]. Yield reductions of up to 61% have been reported in soybeans (*Glycine max*) and 50–60% in wheat (*Triticum aestivum*) [43].

### 2.2. Plant Growth Condition

The study was conducted on *Chenopodium album* L. plants. The seeds used were collected in 2019 from non-cultivated fields located in the southern part of Gliwice, Poland. Throughout the experimental period, plants were grown under ex situ conditions in a greenhouse at a constant temperature of 22 °C. Natural light was supplemented by lamp illumination under long-day conditions (16 h). Plants used for the experiments were obtained from seeds sown directly in a box filled with substrate (GO M11 Seeding Compost, Jiffy Products International B.V.) and covered with a layer of clear plastic wrap. One week after germination, seedlings were transplanted into pots with the same substrate, watered with tap water, and grown for another two weeks until the start of the experiment. Three-week-old plants were sprayed with the tested formulations and watered with tap water if needed. The effect of the treatments was observed depending on the experiment—10 days after spraying for plants treated with dioxolanes alone, and 7 and 21 days after spraying for plants treated with herbicide alone or with the addition of dioxolanes.

### 2.3. Experimental Design, Treatments, and Evaluation

#### 2.3.1. Effect of 1,3-Dioxolanes on Plants

The first part of the experiment examined the effects of 1,3-dioxolanes on plants. To compare the results obtained, a reference set (the so-called control sprayed with purified water), and three tested sets treated with 1, 5, or 10% *v*/*v* solution of DMD (Sigma-Aldrich, St. Louis, MO, USA), TMD (Alfa Aesar, Haverhill, MA, USA), and DDM (Sigma-Aldrich). The prepared solutions with a total volume of 5 mL were applied once using a fine-drop sprayer, on a sample of four plants. Ten days after spraying, the condition of the plants was visually assessed, and parameters such as photosynthetic apparatus efficiency and hydrogen peroxide (H_2_O_2_) content were measured for each plant in the set.

The visual evaluation was carried out for *C. album* based on photographic documentation done with a Sony Alpha 350 camera. Hydrogen peroxide (H_2_O_2_) content was determined using a spectrophotometric method, according to Rudnicka et al. [44], with all the modifications described there. The H_2_O_2_ concentration was expressed as μmol/g fresh weight.

Tests of photosynthetic apparatus efficiency based on ChlF are widely used to analyze the effects of abiotic and biotic stresses on plant health [45,46]. Thus, ChlF parameters were measured using an OS30p+ device (Opti Sciences Inc., Hudson, NH, USA) for the youngest, fully developed, previously dark-adapted leaves, exposed to a saturating pulse of red light (3500 μmol photons × m^−2^ × s^−1^). The following parameters were selected to evaluate the effect of 1,3-dioxolanes: initial (F_O_) and maximum (F_M_) fluorescence, time required to reach maximum fluorescence (T_FM_), maximum quantum yield of primary photochemistry of photosystem II—the ratio of variable fluorescence (F_V_) to F_M_ − F_V_/F_M_, and area above ChlF induction curve (A_M_).

#### 2.3.2. Evaluation of the Efficacy of Dioxolane-Assisted Herbicide

First, the response of *C. album* to a range of reduced herbicide doses was evaluated. Second, the effect of applying the herbicide together with one of the 1,3-dioxolanes was studied. In both cases, plant survival and rapid induction of ChlF were evaluated (OJIP test).

Plants were treated three weeks after germination with aqueous solutions of Round-Up^®^ 360 Plus (Bayer AG) herbicide at various concentrations without or with the addition of 1,3-dioxolanes. Doses tested included 0 mg/L, 72 mg/L, 144 mg/L, 288 mg/L, 360 mg/L, 432 mg/L, 576 mg/L, or 720 mg/L of herbicide and 1 vol.% DMD, TMD, DMD, or MIX (DMD, TMD, and DDM in equal proportions). Samples containing six plants were sprayed with 5 mL of the solution. Table A1 shows the conversions of the contents (mg) and concentrations (mg/L) of the pure glyphosate used. The recommended dose for *C. album* is 1800 mg/L. However, previous herbicide efficacy studies have shown that doses greater than 720 mg/L result in complete plant death under ex situ conditions. The volume of 5 mL was determined by estimating the amount of liquid recommended for the area (45 × 45 cm) occupied by six plants in the experiment.

A rapid ChlF induction test (OJIP test) was performed for each plant in both parts of the experiment 7 days after spraying. Leaf sections were adapted to darkness for at least 30 min before measurement. ChlF was recorded on the upper surface of three fully developed leaves using an OS30p+ device (Opti Sciences Inc., Hudson, NH, USA). The ChlF was induced by red light of 3500 μmol photons × m^−2^ × s^−1^ and fluorescence signals were recorded in a time scan to 1 s after the onset of irradiation. Plant mortality was evaluated 21 days after treatment by counting dead plants.

### 2.4. Statistical Analysis

Data obtained was analyzed using Statistica, version 12 (Statsoft Inc, Tulsa, OK, USA, 2013). The assumptions of normality distribution and homogeneity of variance were ensured. For the statistical analyses, the *t*-test was used to compare the values of the control and test samples. An alpha level of 0.05 was used as the cut-off point for significance [47].

## 3. Results

### 3.1. Effect of 1,3-Dioxolane on Chenopodium Album

Visual evaluation of the plants (Figure A1, Figure A2 and Figure A3) shows that the plants sprayed with 1% and 5% solution did not differ physiologically and morphologically from those sprayed with water. Changes were observed only at the highest concentration, i.e., a slight reduction in leaf area or a slight yellowing. Statistical analysis of ChlF parameters (F_O_, F_M_, T_FM_, S_M_, F_V_/F_M_, A_M_) for the concentrations of 1% and 5% 1,3-dioxolane showed no significant differences from control values. The highest concentration of 10% 1,3-dioxolanes resulted in a statistically significant change compared to the control (see Figure A4). For TMD, there was a decrease in F_O_ and for DMD, there was an increase in the A_M_ parameter. The average H_2_O_2_ content in *C. album* leaves treated with all concentrations of 1,3-dioxolanes was lower than in control plants.

### 3.2. Effect of Glyphosate on ChlF Induction Curve

To determine the survival rate and to study the effect of glyphosate doses on the OJIP ChlF kinetics, *C. album* was sprayed with a commercial formulation of RoundUp^®^ 360 Plus herbicide at various concentrations containing 0 mg/L, 72 mg/L, 144 mg/L, 288 mg/L, 360 mg/L, 432 mg/L, 576 mg/L, or 720 mg/L of herbicide. At the time when the plants were still in good condition, ChlF measurements were carried out (7 days after spraying). ChlF was measured for both the plants from the control sample (0 mg/L) and all other samples tested (Figure 2A), changes in OJIP fluorescence rise kinetics after herbicide treatments were calculated as the variable fluorescence curves (ΔV_t_) (1), using the control plants as the reference (Figure 2B), according to Hassannejad et al. 2020 [38]. The curves are presented as average fluorescence values expressed in arbitrary units [a.u.] plotted on a logarithmic time scale.
(1)$${\Delta V}_{t}=\left(F_{t}-{\;F}_{O}\right)/\left(F_{M}-{\;F}_{M}\right)-{\;V}_{\mathrm{tREF}}$$

For detailed analysis, several OJIP test parameters were selected (Figure 3 and Figure 4): initial fluorescence (F_O_), the fluorescence value at several time points on the curve (F_20 μs_, F_300 μs_, F_2 ms_, F_30 ms_), the maximal fluorescence (F_M_), the time required to reach maximal fluorescence (T_FM_), the normalized total surface area over the OJIP curve (S_M_), the maximum photochemical yield of PSII (F_V_/F_M_), and the maximum water splitting efficiency on the donor side of PSII (F_V_/F_O_).

It can be concluded that the parameters: F_O_, F_20 μs_, F_300 μs_, and F_2 ms_ increased with increasing glyphosate concentration in the solution used for spraying. Statistically significant differences for these four parameters were observed at a glyphosate concentration of 432 mg/L (Figure 4, differences for at F_300 μs_ and F_2 ms_). At the highest glyphosate concentration of 720 mg/L (Figure 4), differences were observed for all four parameters. The other parameters of the OJIP test (F_30 ms_, F_M_, S_M_, F_V_/F_M_, F_V_/F_O_) decreased with increasing glyphosate concentration.

ChlF measurements of samples treated with the herbicide alone showed the typical polyphase curve of fluorescence induction, characteristic of glyphosate in the OJIP test (Figure 2A) [38,48]. The results obtained in the present study for *C. album* showed a relationship between the change in the kinetics of the fluorescence induction curve and the dose of glyphosate used. With increasing glyphosate concentration, a tendency for the ChlF induction curve to flatten was observed, with the curve tending toward higher fluorescence values in the latter part of the O-J phase and a marked decrease in fluorescence values in the I-P phase, resulting in a decrease in maximum fluorescence (F_M_).

The relative ΔV_t_ curves indicate that glyphosate treatment resulted in damage to the oxygen-evolving complex (OEC) and excessive reduction of quinone A [38,48]. A clear deflection of the curve shows this in the O-J phase at the K point. The decrease in the F_V_/F_O_ parameter with increasing glyphosate concentration confirms the OEC damage. Looking at the further trajectory of the ΔV_t_ curves, we note that they reach another peak in the J–I phase, reflecting the excessive reduction in electron transporters such as the secondary electron acceptor, plastoquinone, cytochrome, and plastocyanin. Also striking is a small peak in the I–P phase, independent of glyphosate concentration, indicating an increased reduction in electron transporters on the acceptor side of PSI, such as ferredoxin or other intermediate acceptors, which may indicate reduced activity or damage to terminal electron acceptors in PSII, including ferredoxin—NADP^+^ reductase.

### 3.3. Effect of Dioxolane-Assisted Glyphosate

The first part of the results shows the effect of dioxolane-assisted glyphosate, assessed by the degree of plant death. After treating plants with serial concentrations of herbicide with or without 1 vol.% 1,3-dioxolane, plant death was evaluated after 21 days. Plants that died in each experiment were counted and survival curves of *C. album* were plotted as a function of herbicide concentration and the addition of 1,3-dioxolane (Figure 5).

When analyzing the survival curve of *C. album* as a function of herbicide concentration, no plant death was observed at glyphosate concentrations of 0–288 mg/L. In the range of 288–360 mg/L, there was a sharp decrease in the survival rate to 50%, after which the survival curve shows a lower slope at concentrations > 360 mg/L and reaches 0% at concentrations of 720 mg/L—effective dose (ED).

The possibility of using 1,3-dioxolanes as adjuvants was evidenced by the reduced survival of *C. album* treated with herbicide containing these compounds. DMD can be considered the best adjuvant at low glyphosate concentrations (from 144–288 mg/L). The data collected show that the addition of DMD to the herbicide at low concentrations increased mortality compared to this analogous concentration of glyphosate alone as well as to glyphosate with the other 1,3-dioxolanes. When TMD was added to the herbicide, a mortality rate of 100% was achieved at the lowest glyphosate concentration (360 mg/L) compared to the other 1,3-dioxolanes and their combination (432 mg/L) and the herbicide alone (720 mg/L).

The second part of the results presents the main objective of the experiment, which was to determine a faster method to evaluate whether the addition of a potential adjuvant to herbicide is more effective against *Chenopodium album*. This was done by analyzing the fluorescence of ChlF long before the appearance of visible symptoms.

The effects of 1,3-dioxolane addition on the OJIP curve and selected ChlF parameters were studied for glyphosate concentrations that caused statistically significant differences between treated and control plants (72–360 mg/L). ChlF induction and relative fluorescence variable (ΔV_t_) curves were plotted for each sample. Detailed analysis of fluorescence parameters with statistical analysis is summarized in radar charts. Data for the 288 mg/L concentration are shown in Figure 6, Figure 7 and Figure 8; other concentrations are listed in the Appendix A (Figure A5, Figure A6, Figure A7, Figure A8, Figure A9 and Figure A10).

At a concentration of 72 mg/L, there were no significant differences between the plants treated with herbicide alone and the other samples treated with herbicide with the addition of 1 vol% 1,3-dioxolanes. Differences in the ChlF induction curves between samples were not significant—the values of the relative fluorescence variable (ΔV_t_) were low (<± 0.06) (Figure A5B). No statistically significant differences were found between ChlF parameters (Figure A6).

Corresponding observations were confirmed for concentrations of 144 mg/L. The values of ΔV_t_ were mostly low (<±0.09) (Figure A7B), except for the herbicide with TMD, where a flattening of the ChlF induction curve was observed in the I-P phase (Figure A7A). It is noteworthy that the herbicide in combination with DMD resulted in statistically significant differences between the ChlF parameters (Figure A8). The parameters F_O_ and S_M_ increased compared to the herbicide alone, while the parameters F_2 ms_, F_V_ /F_M_, and F_V_/F_O_ decreased. Higher mortality is observed with this combination than with the herbicide alone (Figure 5).

The greatest differences in the parameters studied between the use of herbicide alone and herbicide with 1,3-dioxolanes were observed at a concentration of 288 mg/L glyphosate (Figure 6 and Figure 7). The values of ΔV_t_ were significant (<± 0.20), and fluorescence parameters F_20 μs_, F_300 μs_, F_2 ms_, and F_30 ms_ reached higher values. The differences in these parameters are statistically significant. The herbicide alone did not change the survival rate of quinoa at this concentration, only in combination with dioxolane an increase in mortality was observed, indicating their effectiveness (Figure 5).

When the glyphosate concentration was increased to 360 mg/L, the effect of dioxolane addition on *C. album* mortality was not observed as strongly, most likely due to reaching the herbicide concentration that caused half of the weed population to die. However, the stimulatory effect of added dioxolane on the herbicide effect was still evident (Figure A9 and Figure A10), but no longer as strong as at the glyphosate concentration of 288 mg/L.

## 4. Discussion

Despite many objections, pesticides continue to have a major impact on agricultural production around the world and are relied upon to protect crops [6]. It is highly likely that agriculture and related intensive food production will not be able to completely eliminate the use of agrochemicals in the coming years. Therefore, alternative approaches to chemical use have been sought in recent years, but for the time it takes to find the golden mean, it may be promising to introduce techniques that allow the precise application of agrochemicals [3]. In many cases, the combined use of herbicides together with adjuvants allows reducing the application rate and limiting the risk of harmful residues entering the environment [50].

The properties of 2,2-dimethyl-1,3-dioxolane (DMD), 2,2,4-trimethyl-1,3-dioxolane (TMD), and (2,2-dimethyl-1,3-dioxan-4-yl)methanol (DDM) enable them to be used as adjuvants added to foliar-applied herbicides. Given the complexity of the process of herbicide uptake by aboveground plant parts, an ideal adjuvant should increase the contact area of the droplet with the plant surface, keep the active ingredients in soluble form, and increase the permeability of the cuticle and plasma membranes. Among the currently used adjuvants, it is difficult to find one that has all these properties [16]. Surfactants mainly provide increased droplet binding. Mineral adjuvants such as ammonium salts, which trap cations that can react with the active ingredient, only affect stability and prevent the formation of insoluble forms of the herbicide. Oil-based adjuvants slow the evaporation of the active ingredient and improve its penetration, e.g., by changing the components of the cuticle, but usually cause foaming of the active ingredient, which is an unfavorable phenomenon [16]. Selected 1,3-dioxolanes allow significant enhancement of droplet retention of solutions of various leaf-applied active ingredients and can directly affect the process of absorption of biologically active compounds such as IAA or herbicides through the cuticular barrier of leaves [35,37].

The final content of adjuvants in ready-to-use pesticide formulations is often more than 50% of the total product and often accounts for most of the price of the formulation [27]. In this case, the application of DMD, TMD, and DDM at a concentration of 1 vol.% was sufficient to significantly enhance the effect of RoundUp^®^ 360 Plus.

The OJIP test provides a good method for examining the early response of *Chenopodium album* to glyphosate. According to several authors [38,51,52,53,54,55], the OJIP test can be used to determine the effects of low doses of herbicide on treated plants. They suggest that it can also be used in studies on glyphosate, although the mechanism of action does not directly involve photosystem II. The results obtained in the present study for *C. album* showed a correlation between the change in ChlF kinetics and the applied dose of glyphosate (Section 3.2). The differences in OJIP kinetics allow the effect of different doses of herbicide to be measured as they are an indicator of the amount of glyphosate that has reached the site of action. The variable relative fluorescence (ΔV_t_) curves show the occurrence of a peak at several steps of electron transport through the electron transport chain. When analyzing the ChlF induction curve, the decrease in the fluorescence maximum and the increase in the fluorescence value in the O-J phase, especially at the K point, are particularly noticeable with increasing glyphosate dose [56]. Nevertheless, this O-J phase appears to be useful in testing new adjuvants for glyphosate, as are other parts of the curve for various herbicides [38,48].

## 5. Conclusions

Each of the 1,3-dioxolanes tested had a positive effect on the efficacy of Roundup^®^ 360 Plus, i.e., improved its herbicidal activity. It was found (Section 3.3) that it is possible to reduce the amount of glyphosate applied and achieve 100% efficacy of this product at a much lower concentration when 1 vol.% DMD, TMD, DDM, or a mixture of these compounds is added. Glyphosate alone did not cause plant death at concentrations up to 288 mg/L. It only reached 100% efficacy at an effective dose (ED) of 720 mg/L. However, the addition of the test compounds resulted in a 40% reduction in the effective dose for DMD, 50% for TMD, and 40% for DDM (Figure 5). The results obtained in the present study show that at lower glyphosate concentrations, the addition of 1% DMD by volume is most effective. No synergistic effect was observed in the MIX, i.e., the mixture of dioxolanes does not increase the effect of the herbicide and acts at the same level as a single dioxolane.

The obtained results indicate that the studied compounds can point the way to minimize the doses of herbicides, as well as other agrochemicals, applied through the foliage, which would lead to a reduction in pollutants introduced into the environment. The concept of using 1,3-dioxolanes to reduce herbicide use is valid, although it will be critical to conduct field studies to verify the results obtained under laboratory conditions.

## Figures and Tables

**Figure 1 cells-12-00777-f001:**
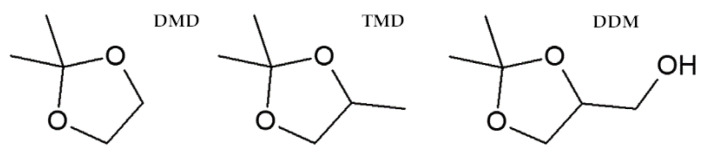
The chemical structures of 2,2-dimethyl-1,3-dioxolane (DMD), 2,2,4-trimethyl-1,3-dioxolane (TMD), and (2,2-dimethyl-1,3-dioxan-4-yl)methanol (DDM).

**Figure 2 cells-12-00777-f002:**
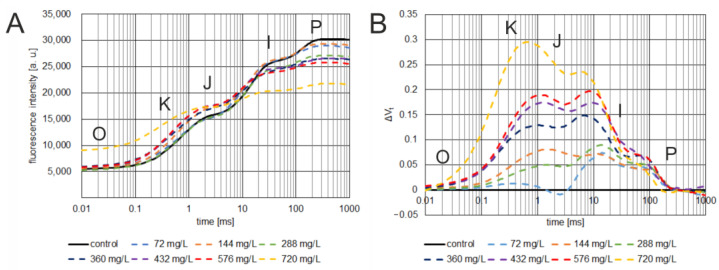
The effect of spraying different concentrations of glyphosate on the ChlF induction curve (**A**) and the relative variable fluorescence induction transients (ΔV_t_) (**B**) of *C. album* leaves 7 days after treatment. The steps of fluorescence rise kinetics are indicated by the letters O, K, J, I, P [48,49].

**Figure 3 cells-12-00777-f003:**
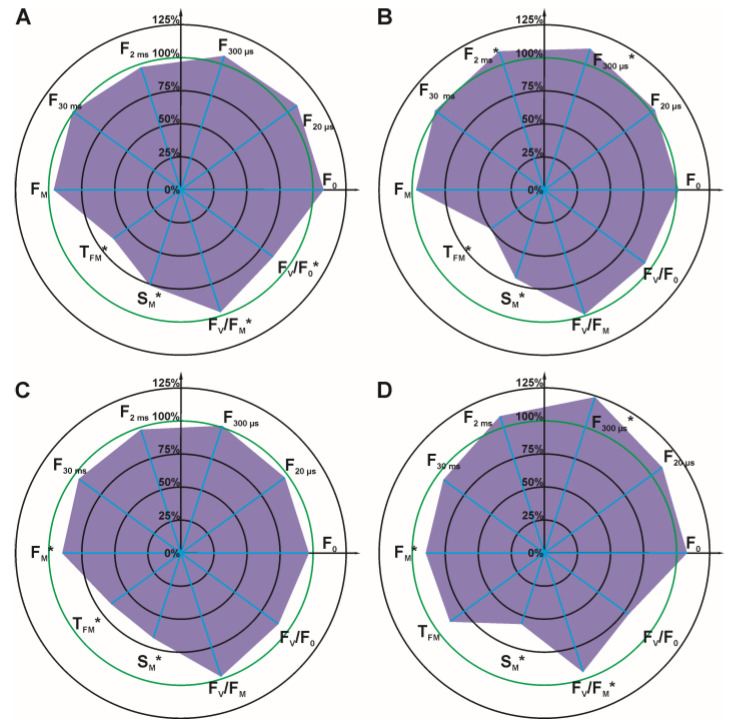
Radar charts of percentage chlorophyll *a* fluorescence parameters of *C. album* leaves 7 days after spraying with glyphosate at 72 mg/L (**A**), 144 mg/L (**B**), 288 mg/L (**C**), 360 mg/L (**D**). The parameters that were measured for the control were arbitrarily set to 100%. Accordingly, each parameter value is shown as a relative value to control. Statistically significant differences in parameters are marked with an asterisk.

**Figure 4 cells-12-00777-f004:**
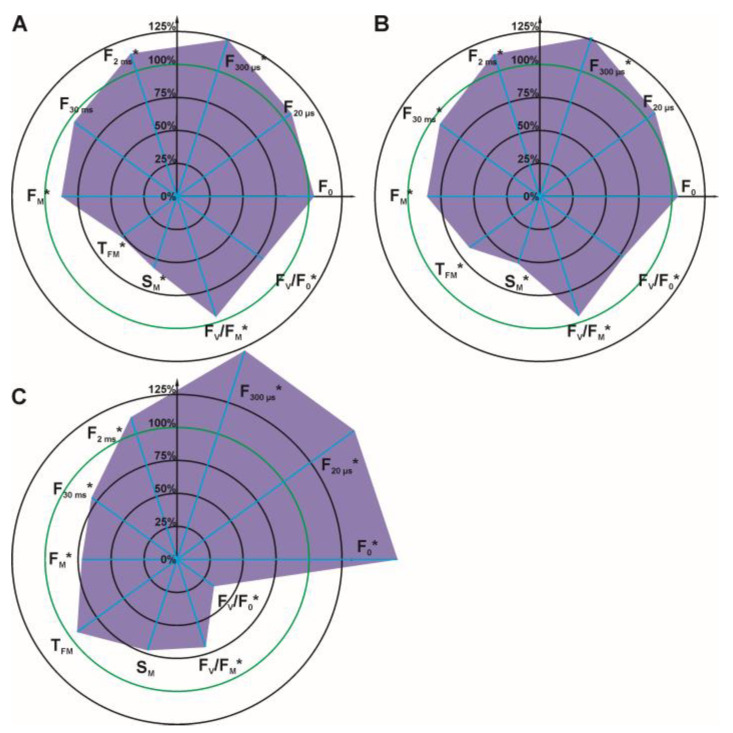
Radar charts of percentage chlorophyll *a* fluorescence parameters of *C. album* leaves 7 days after spraying with glyphosate at 432 mg/L (**A**), 576 mg/L (**B**), 720 mg/L (**C**). The parameters that were measured for the control were arbitrarily set to 100%. Accordingly, each parameter value is shown as a relative value to control. Values of the parameters for the control plants were taken as 100% and marked with a green circle on the graphs. Statistically significant differences in parameters are marked with an asterisk.

**Figure 5 cells-12-00777-f005:**
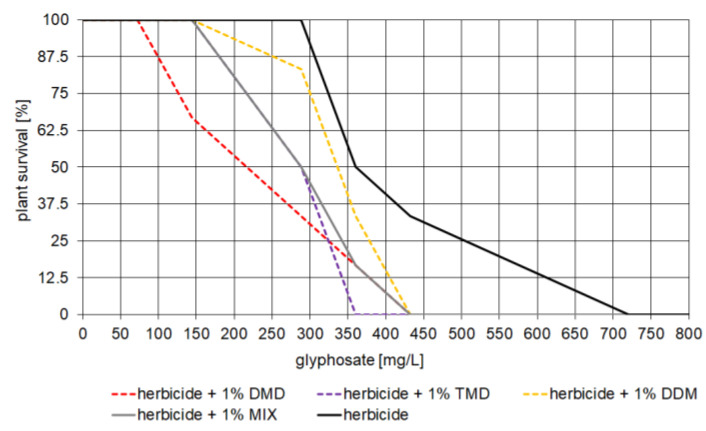
Survival curves of *Chenopodium album* as a function of herbicide concentration and addition of 1 vol.% 1,3-dioxolane 21 days after treatment with the different test variants.

**Figure 6 cells-12-00777-f006:**
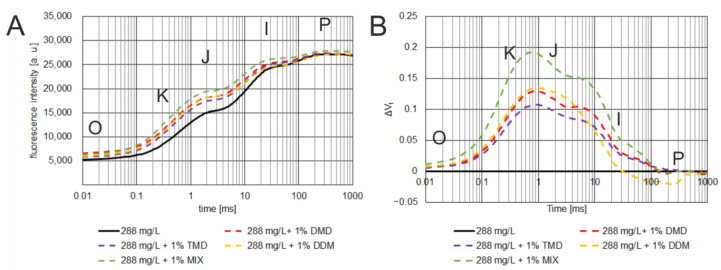
The effect of spraying glyphosate at a concentration of 288 mg/L, with or without the addition of 1 vol.% 1,3-dioxolanes, on the chlorophyll *a* fluorescence induction curve (**A**) and the relative variable fluorescence induction transients (ΔV_t_) (**B**) of *C. album* leaves 7 days after treatment. The curves show average fluorescence values expressed in arbitrary units [a.u.] as a logarithmic function of time. The following phases of chlorophyll fluorescence induction are denoted by the letters O, K, J, I, P [48,49].

**Figure 7 cells-12-00777-f007:**
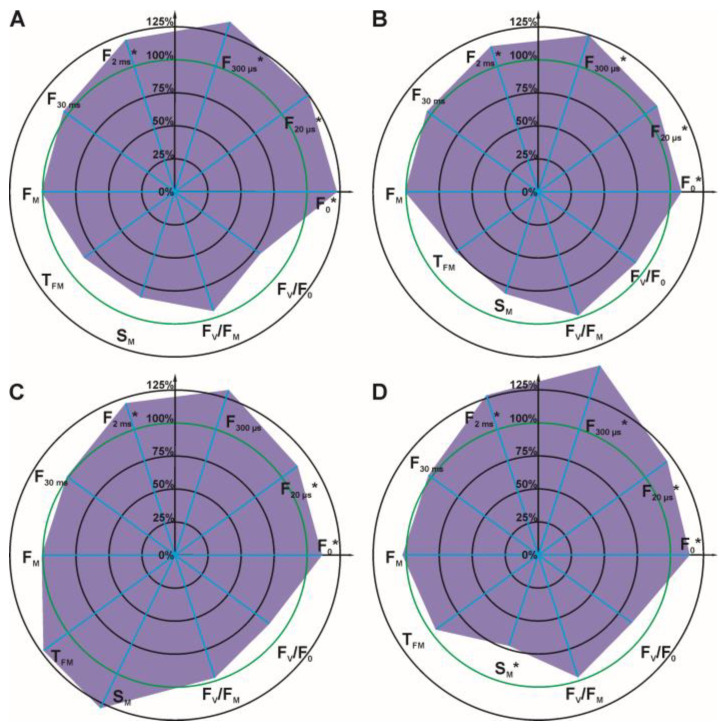
Radar charts of percentage chlorophyll *a* fluorescence parameters of *C. album* leaves after spraying with an agent containing glyphosate at 288 mg/L with 1 vol.% of DMD (**A**), TMD (**B**), DDM (**C**), or MIX (**D**). The parameters that were measured for the control were arbitrarily set to 100%. Accordingly, each parameter value is shown as a relative value to control. Values of the parameters for the control plants were taken as 100% and marked with a green circle on the graphs. Statistically significant differences in parameters are marked with an asterisk.

**Figure 8 cells-12-00777-f008:**
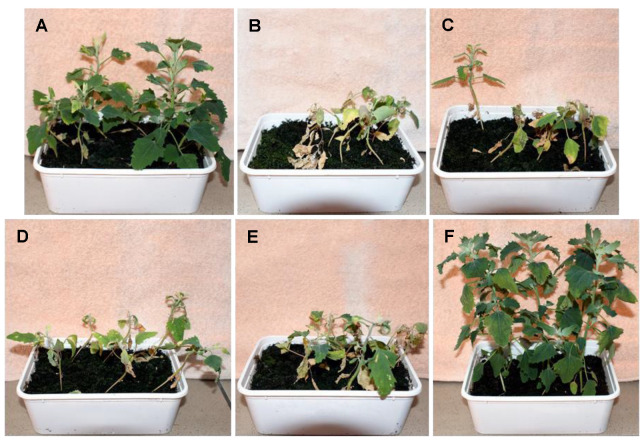
Plants 21 days after treatment with glyphosate at a concentration of 288 mg/L. (**A**), with the addition of 1% DMD (**B**), 1% TMD (**C**), 1% DDM (**D**), and 1% MIX (**E**) compared to plants treated with water (**F**).

## Data Availability

All data generated or analyzed during this study are included in this published article and its Appendix A or are available from the corresponding author upon reasonable request.

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
