# Peer review of "Potential Role of Low-Molecular-Weight Dioxolanes as Adjuvants for Glyphosate-Based Herbicides Using Photosystem II as an Early Post-Treatment Determinant"

_cells, 2023, doi:10.3390/cells12050777_

Round 1

Reviewer 1 Report (Previous Reviewer 1)

The authors have addressed all the concerns I raised during my first review and this has greatly improved the overall flow and ease of understanding of the manuscript. Therefore, it can be accepted for publication in its current form.

Reviewer 2 Report (Previous Reviewer 2)

The manuscript is changed according to the comments

This manuscript is a resubmission of an earlier submission. The following is a list of the peer review reports and author responses from that submission.

Round 1

Reviewer 1 Report

General comments

The paper “Potential role of low-molecular-weight dioxolanes as adjuvants for glyphosate-based herbicides using photosystem II as an early post-treatment determinant” presents data to support improved efficacy of herbicides (glyphosate) when applied together with an adjuvant such as 1,3-dioxolanes. The use of herbicides in agriculture has turned out to be a necessary evil with both beneficial effects for farmers but negative effects on the environment. Therefore, technologies are needed to replace herbicides or improve their efficiency so that less of it can be used for the same effects. The data provided in the paper clearly shows the benefit of the glyphosate+adjuvant combination which can be adapted to benefit farmers and the environment. The data presented also shows that even low concentrations of the adjuvants when combined with herbicide can be effective against weeds. The paper is well-written and makes for an interesting read. However, there are some major and minor flaws that render the paper unpublishable in its current state. Nevertheless, these major, as well as minor issues, can be addressed to allow the paper to be accepted for publication. These are outlined below:

Major corrections:

Results:

P7L277: The authors indicate that 'plant death was assessed after 14 days'. However, the title of the corresponding figure shows that the data is from 21 days after treatment. This disparity needs to be reconciled.

didn’t provide any discussion of their work. They need to have a

Discussion

The authors didn’t provide any discussion of their work. They need to have a separate section for ‘discussions’ where they discuss the meaning and implications of their findings in relation to the existing literature as indicated in the instructions to authors.

Minor comments:

The short form of Chenopodium album should be C. album and not ‘Ch. album’.

Methods:

P3L118: The English or common name of the weed should be stated in parenthesis after the scientific name for easier reference to it without confusing the reader.

P4L155 and L167: To make it easier to reconcile the results with the ‘Materials and methods’, the glyphosate treatments should be stated as the concentration (in mg/L) rather than stating it as the volume. The Table in the appendix (supplementary materials) can still be maintained for reference and conversion purposes.

P4L159, P5L199: The authors refer to C. album as ‘white quinoa’. However, the correct English name is ‘white goosefoot’.

P5L204: What was the basis for using the non-parametric t-test? The authors should state their reason for this choice in the text.

Results:

Figures should be numbered properly and consistently in the main text and in the ‘appendix’. For example, in L211, the authors make mention of Appendix Fig. 1-3. However, in the appendix, the figures are numbered as Figure A1, A2, and A3. This can be a source of confusion for readers. Therefore, all figures should be numbered consistently. Some of the figures in the appendix are numbered as ‘Figure A4’ whereas others are numbered ‘Figure S5’. I suggest the author rename the ‘Appendix’ as ‘Supplementary Materials’ and stick with figure and table naming convention which precedes the figure/table number with an ‘S’. For example, ‘Figure S1’ ‘Table S1’ etc.

Some of the information in figures 1,4 and 5, such as the axis labels and figure legends are in a different language (Polish?). These need to be translated into English. The same should be done for some information in the table and figures in the ‘Appendix’.

P5L223-224: The actual concentrations of glyphosate should be indicated since this is what has been provided in the figure legends. The volumes of glyphosate may be put in parentheses for reference purposes.

P5L230: The meaning of parameters appearing in the equation should be made clear, especially those that have not been introduced previously.

P7L277: The authors indicate that 'plant death was assessed after 14 days'. However, the title of the corresponding figure shows that the data is from 21 days after treatment. This disparity needs to be reconciled.

Figures 1,2,3,5 and 6: The authors should indicate the time after the application of the treatment during which the measurements were made. This should also be done for other figures in the 'Appendix'.

Reviewer 2 Report

The present study investigated the effects of low-molecular-weight dioxolanes (DMD, TMD and DDM) on the efficacy of commercial pesticide (RoundUp® 360 Plus) at low doses.

The idea of this study is very interesting and practically significant, but the presentation of the results of this study does not meet the requirements of a high-level journal. Moreover, there is no discussion of the presented results in the manuscript, so the reader is put in a difficult position when he himself has to analyze the response of studied parameters to influencing factors, such as herbicide, low-molecular-weight dioxolanes, and their combined action. The results of this study consist mainly of plurality of chlorophyll a fluorescence parameters interchangeable with each other. However, a large amount of data on chlorophyll fluorescence confuses the author rather than clarifies the issue of the effect of the studied substances on weeds. The conclusion made by the authors based on the results of this study deserves scientific attention, it is logical and practically significant, but it can be drawn based more on the data on plant death presented in the Fig. 4 than on chlorophyll fluorescence data.

The FV/FM parameter of chlorophyll fluorescence is the most widely used parameter reflecting damage to the photosynthetic apparatus of plants. The results of this study showed no effect of low-molecular-weight dioxolanes on this parameter (Fig. 6). At the same time, in Fig. 7 shows seedlings at various stages of damage. The discrepancy between the state of seedlings and the FV/FM parameter, unfortunately, is not the only one in this work.

Figure A1-A3 show that within each treatment of the four plants, the plants vary greatly in the height and other growth parameters, i.e. the sample is not aligned. This may be one of the reasons for the insignificant response of the studied parameters to any impact.

Minor comments:

Abstract does not reflect the results of the study;

Line 93: What DMD meanes?

Line 92, 93: (γ) and (θ) are not necessary because the authors don’t use them in the text;

Line 97: What logP means?

Please, formulate a hypothesis, aim and objectives of this study;

Line 135-138: When the experiment ended? After three weeks? Seedlings not watered after three weeks? Please clarify this;

Line 150: The data of leaf H2O2 content not shown below and the method of  measurement of H2O2 not shown too;

Line 182, 192: Why different equipment was used for Chl a measurements?

Line 222: That LD50 and LD100 mean? How they were measured?

Line 223, 244, 250, 284, Fig. 1 and so on: Please, use uniform for RoundUp content - mL or mg/l;

Fig. 1: Please, clarify the phases O,K,J,I,P or give a reference explaining this;

Fig 2, 3, 6 are hard to read. Please, show this data to real values, not to %. Use a table;

Lines 262-274: Please, give the references for each sentence;

Fig 1, 4, 5 should be translated into English;

Best regards,